# Effects of *Crocus sativus* L. Floral Bio-Residues Related to Skin Protection

**DOI:** 10.3390/antiox13030358

**Published:** 2024-03-17

**Authors:** Nuria Acero, Dolores Muñoz-Mingarro, Ana Gradillas

**Affiliations:** 1Departamento de CC Farmacéuticas y de la Salud, Facultad de Farmacia, Universidad San Pablo-CEU, CEU Universities, Urbanización Montepríncipe, 28660 Boadilla del Monte, Spain; 2Departamento de Química y Bioquímica, Facultad de Farmacia, Universidad San Pablo-CEU, CEU Universities, Urbanización Montepríncipe, 28660 Boadilla del Monte, Spain; dmumin@ceu.es; 3Centro de Metabolómica y Bioanálisis (CEMBIO), Facultad de Farmacia, Universidad San Pablo-CEU, CEU Universities, Urbanización Montepríncipe, 28660 Boadilla del Monte, Spain; gradini@ceu.es

**Keywords:** *Crocus sativus* L., saffron bio-residues, phenolic compounds, antiaging, dermo-protection, antioxidant capacity

## Abstract

The cultivation of *Crocus sativus* L. to obtain the saffron spice generates a large amount of biowaste, constituted mainly by the flower’s tepals. The aim of this work was to evaluate the antioxidant and dermo-protective effect of a complex methanolic extract of *C. sativus* tepals. The extract’s major phenolic content was analyzed using ultra-high performance liquid chromatography with electrospray ionization, coupled with quadrupole-time-of-flight-mass spectrometry (UHPLC-ESI-QTOF-MS). Then, the antioxidant in vitro activity of the extract was studied and related to their chemical composition. Likewise, the effect on intracellular ROS levels in HepG2 and Hs27 cell culture was determined in normal culture and under hydrogen-peroxide-induced oxidative stress. Finally, tyrosinase, hyaluronidase, collagenase, elastase, and xanthine oxidase assays were carried out to determine the dermo-protective capacity of the extract. The high polyphenol content, including flavonoids and anthocyanins, explains the antioxidant effect of the extract both in vitro and in culture assays. The extract has a significant and remarkable protective capacity against oxidative stress induced in culture of the two studied cell lines. It is also remarkable in its ability to inhibit hyaluronidase, tyrosinase, and xanthine oxidase. Results pointed out this biowaste extract as a promising ingredient in the composition of cosmetics.

## 1. Introduction

The skin is the largest organ of the human body which constitutes a protective barrier between internal and external environment. Over the years, there is a natural skin aging known as intrinsic aging that depends on genetic factors and body changes. In addition, exposure to external elements results in an extrinsic aging, which is mainly due to UV radiation in the rage of 320–400 nm, which is known as photoaging [1]. Because of continued exposure to UV rays, hydroxyl, hydrogen peroxide, singlet oxygen, and superoxide radicals are generated, leading to cellular damage. This oxidative stress induces loss of skin elasticity and, therefore, wrinkles, laxity, pigmentation, actinic purpura, brown spots, actinic elastosis, and lesions that can result in different types of skin cancer [2].

As a result of aging, levels of elastin, collagen, and hyaluronic acid decrease together with an increase in some enzymes’ activity such as collagenase, hyaluronidase, and elastase. Therefore, a deterioration of dermal extracellular matrix occurs [3]. Collagen and elastin confer tensile strength and recoil properties, respectively, keeping skin elastic, youthful, and healthy [4]. Another major component of the skin extracellular matrix is hyaluronic acid, which plays an important role in wound healing and tissue repair processes. This polymer maintains skin moisture levels as it can bond water to tissues [5]. During aging, hyaluronidase is synthesized, and levels of hyaluronic acid decrease, and hence, skin smoothness, emollience, and youngness diminish [6].

Photoaging also involves the accumulation of melanin whose excess and abnormal distribution causes the appearance of dark spots. Melanin synthesis occurs in melanocytes catalyzed by tyrosinase [7]. In this sense, the inhibition or downregulation of this enzyme prevents pigmentation disorders, one of the goals of skin antiaging agents [8].

The search for new antiaging compounds mainly from natural origin has gained attention during the last few decades and is still expanding. In addition, the skin care market based on botanicals continues to see strong growth. Furthermore, antiaging and antiwrinkle cosmetics are frequently supplemented with antioxidants for their ability to prevent and treat skin photo-damage [9]. Having these in consideration, skin treatment with natural compounds could reduce photoaging, not only scavenging free radicals, but also inhibiting the expression or activity of the mentioned enzymes [10].

Saffron cultivation (*Crocus sativus* L., Iridaceae) involves a high manual labor requirement because the process is not machined. The collection of the flower in the field and the subsequent separation of stigmas are totally by hand and therefore, are costly tasks. After manual flower gathering, stigmas represent just 7.4% of the total flower weight, generating 92.6 g of biowaste per 100 g of flowers [11]. In this sense, tons of bio-residues are generated each year because of saffron cultivation and production. This biowaste is mainly made up of the flower tepals (flower perianth where petals and sepals are not clearly differentiated). The interest in new potential commercial values for these bio-residues is increasing. The chemical composition of several extracts from this *C. sativus* flower tepals has been reported, demonstrating a high phenolic content [12]. Studies confirm that saffron tepals are not waste products; on the contrary, they are a promising source of flavonoids and other interesting natural products that could act synergistically and which have shown potential pharmacological applications [11,12,13,14,15,16,17].

The aim of this work was to obtain a complex extract of *C. sativus* tepals, preventing degradation of anthocyanins and other phenolic compounds along the process. To this end, methanol plus 0.1% of HCl was used as solvent, and low temperatures were kept during obtention and concentration of the extract. Moreover, determination of the polyphenolic profile and antioxidant capacity was carried out. Towards this purpose, the methanolic extract was analyzed using ultra-high performance liquid chromatography with electrospray ionization, coupled to quadrupole-time-of-flight-mass spectrometry (UHPLC-ESI(±)-QTOF-MS), providing a comprehensive characterization of main phenolic content.

Additionally, the antioxidant activity of the methanolic extract was analyzed and related to their chemical composition. In this respect, the ability of the extract to scavenge different free radicals such as superoxide, hydroxyl, nitrogen, DPPH, or ABTS were evaluated. Likewise, the effect on intracellular ROS levels in HepG2 and Hs27 cell culture was determined in normal culture and under hydrogen-peroxide-induced oxidative stress. Finally, enzymatic assays were carried out to determine the dermo-protective capacity in vitro of the extract, and for this purpose, the effect of the tepal extract on different enzymes was studied: tyrosinase, hyaluronidase, collagenase, elastase, and xanthine oxidase.

## 2. Materials and Methods

### 2.1. Chemicals and Reagents

Purified water was obtained using the Milli-Qplus185 system (Millipore, Billerica, MA, USA). MS grade methanol (MeOH) was purchased from Merck (Darmstadt, Germany). MS grade formic acid (FA) was from Sigma-Aldrich Chemie GmbH (Steinheim, Germany). In general, enzymes, substrates, free radicals, cell culture mediums, and supplements were purchased from Sigma-Aldrich (St. Louis, MO, USA).

### 2.2. Plant Material and Extract Obtention

*C. sativus* plants used in this study came from a traditional organic crop located in Vinaceite (Teruel, Spain). Flowers were hand-harvested in early October 2018 and placed in traditional wicker baskets. After stigmas were hand-picked, the biowaste, mainly tepals, were frozen at −20 °C and sent to the laboratory where they were kept frozen until use. The samples were provided by Mrs Eva Bielsa.

An amount of 30 g of bio-residues was ground with liquid nitrogen using a mortar and extracted with 300 mL of methanol with 0.1% of commercial HCl. Samples were sonicated in an ultrasonic bath for 30 min. They were then filtered, and 300 mL more of methanol with 0.1% HCl was added to perform another 15 min sonication, and then the extract was filtered again. This last step was repeated twice. The obtained methanolic extract was concentrated in a rotavapor at 40 °C (Buchi R-114) and then stored at 4 °C until use. Extract yield was 11.72% (*w/w*). Saffron tepals’ water content was 72.8 ± 1.3%.

### 2.3. Phytochemical Analysis

#### 2.3.1. Total Phenolic, Flavonoid, and Anthocyanin Content

Total phenol content determination of the extract was carried out by a colorimetric method, using a gallic acid calibration [18].

Five μL of different concentrations of the extract and 80 μL of 10% Folin-Ciocalteau reagent were added to each well of a plate of 96, and then, the plaque was stirred. After 5 min, 160 μL of 7.5% sodium carbonate (Na_2_CO_3_) was added, and the plaque was stirred again. After 30 min of incubation in darkness, absorbance was measured into a Spectrostar-Nano (BMG Labtech, Ortenberg, Germany) reader at 765 nm. Quantification was carried out on the basis of a gallic acid standard curve and expressed as the percentage of gallic acid equivalents (GAE). All the assays were performed in triplicate.

Total flavonoid concentration in the extract was determined in triplicate by a colorimetric assay [19]. Epicatechin was used as standard.

An amount of 200 μL of different concentrations of the extract, 200 μL of Milli-Q water as blank, or 200 μL of different concentrations of epicatechin (8 serial dilutions starting from 0.1 mg/mL), were mixed with 800 μL of Mili-Q water, and 60 μL of 5% sodium nitrite (NaNO_2_). After five minutes, 60 μL of 10% aluminium trichloride (AlCl_3_) was added, and the next minute, 400 μL of sodium hydroxide (NaOH) 1 M. Immediately, the appearance of color occurs, reading the absorbances against the blank at 510 nm in Spectrostar-Nano spectrophotometer (BMG Labtech). Results are expressed as epicatechin equivalents (EE) percentage.

The pH differential spectroscopic method was used for the determination of total anthocyanin content [20].

Five hundred μL of a 10 mg/mL extract dilution in methanol with 0.1% commercial HCl were mixed with 7 mL of two different buffers: 0.2M KCl pH 1.0 and 1M CH3COONa pH 4.5. After incubation for 15 min at room temperature, absorbance values were measured at 510 and 700 nm. The extract was analyzed in triplicate.
Abs = [(Abs510 − Abs700)pH1.0 − (Abs510 − Abs700)pH4.5]

Anthocyanin concentration was calculated following the equation:C (mg/L) = [Abs/ε × L] × MW × DF × 103
where ε is delphidin-3-O-glucoside molar extinction coefficient = 29,000 L/mol × cm; MW is delphidin-3-O-glucoside molecular weight = 500.84 g/mol; L is the cuvette optical path length = 1 cm; and DF is the dilution factor.

The total anthocyanin content was expressed as percentage of delphidin-3-O-glucoside equivalents in the extract.

#### 2.3.2. Determination of the Polyphenolic Profile by UHPLC-ESI(±)-QTOF-MS Analysis

The identification of polyphenolic content present in the *C. sativus* tepals sample was performed following a methodology previously developed by the Centre of Metabolomics and Bioanalysis (CEMBIO), based on analysis by UHPLC/ESI-QTOF-MS [20].

For that, a methanolic extraction was performed as follows: 1000 μL of methanol was added to 10 mg of the extract. The mixture was vortexed for 5 min and centrifuged at 10,000× *g* for 5 min at 4 °C.

The supernatants were then collected and transferred to a Chromacol vial (Thermo Fisher Scientific, Madrid, Spain) for LC/MS analysis. The whole procedure was performed in duplicate.

Then, samples were analyzed on an Agilent 1290 Infinity series UHPLC system coupled through an electrospray ionization source (ESI) with jet stream technology to a 6545 iFunnel QTOF-MS system (Agilent Technologies Inc., Waldbronn, Germany).

For the separation, a volume of 2 μL was injected in a reversed-phase column (Zorbax Eclipse XDB-C18 4.6 × 50 mm, 1.8 μm, Agilent Tech.) and kept at 40 °C. The flow rate was 0.5 mL/min with a mobile phase consisting of solvent A: 0.1% formic acid in ultrapure water, and solvent B: methanol. Gradient elution consisted of 2% B (0–6 min), 2–50% B (6–10 min), 50–95% B (11–18 min), 95% B for 2 min (18–20 min), and returned to starting conditions 2% B in one minute (20–21 min) to finally keep the re-equilibration with a total analysis time of 25 min.

Detector was operated in full scan mode (*m*/*z* 50–1500) in positive and negative ESI modes at a scan rate of 1 scan/s. A solution consisting of two reference mass compounds were used throughout the whole analysis: purine (C_5_H_4_N_4_) at *m*/*z* 121.0509 (protonated purine) for the positive and *m*/*z* 119.0363 (deprotonated purine) for the negative ionization modes; and HP-0921 (C_18_H_18_O_6_N_3_P_3_F_24_) at *m*/*z* 922.0098 (protonated HP-921) for the positive and *m*/*z* 966.0007 (HP-0921 + formate) for the negative ionization modes. These masses were continuously infused into the system through an Agilent 1260 Iso Pump at a 1 mL/min (split ratio 1:100) to provide a constant mass correction.

For positive and negative ionization mode, the capillary voltage was ±4000 V. The source temperature was 225 °C. The nebulizer and gas flow rates were 35 psi and 11 L/min, respectively, using a fragmentor voltage of 75 V and a radiofrequency voltage in the octupole (OCT RF Vpp) of 750 V, while 10, 20, and 30 eV collision energies were used for MS/MS runs.

MassHunter Workstation version B.07.00 software (Agilent Technologies Inc.) was used for control and data acquisition and MassHunter Qualitative Analysis software (B.08.00, Agilent Technologies Inc.) was used for extraction and characterization of phenolic compounds.

### 2.4. In Vitro Scavenging Activity

#### 2.4.1. DPPH Assay

The 1,1-diphenyl-2-picridacil (DPPH٠) assay was performed colorimetrically [21]. One hundred μL of sample, standard (ascorbic acid), or methanol as blank, was mixed in a 96-well plate with 100 μL of DPPH 1 mM. The plate was incubated at room temperature and darkness for 20 min. Absorbances were then measured at 517 nm in a plate reader (Spectrostar Nano BMD Labtech). The concentration able to reduce 50% of the DPPH radical (IC_50_) was obtained.

#### 2.4.2. Radical ABTS Scavenging Assay

The ability of *C. sativus* extract to reduce ABTS radical (2,2′ azinobis-(3-ethylbenztiazoline)-6-sulfonic acid) was conducted following Tu and Tawata (2015) [22] protocol. All reagents were prepared in PBS, and the assay was performed in triplicate on a 96-well plate. Each well contained 120 μL of ABTS (400 μM), 3.6 μL of myoglobin (400 μM) with potassium ferrocyanide (740 μM), 6 μL of extract, standard at different concentrations or PBS, 68.4 μL of water. Just before measuring, 102 μL of H_2_O_2_ (450 μM) were added. Absorbances were measured in a Spectrostar Nano plate reader (BMG Labtech) at 734 nm every min for 10 min. The percentage of radical reduction and the IC_50_ was calculated.

#### 2.4.3. Superoxide Anion Scavenging Assay

The xanthine–xanthine oxidase enzyme catalyzes the oxidation of hypoxanthine to xanthine, and of xanthine to uric acid. In this catalysis, a superoxide radical (O_2_^−^) is generated, which can reduce the NBT (nitroblue tetrazolium) [19]. The analysis was carried out in triplicate on a 96-well plate. All reagents were prepared in phosphate buffer (50 mM KH_2_PO_4_/KOH, pH = 7.4). An amount of 62.5 μL of phosphate buffer, 10 μL of EDTA (ethylenediaminetetraacetic acid) (15 mM), 15 μL of hypoxanthin (3 mM), 25 μL of NBT (0.6 mM), 25 μL of different concentrations of the extract, or standard (gallic acid), or phosphate buffer as blank, and 25 μL of xanthine oxidase (1 U/10 mL of buffer) were mixed. Absorbance was measured at intervals of 5 min for 40 min after the addition of the enzyme, at a temperature of 37 °C in a Spectrostar Nano plate reader (BMG Labtech) at 560 nm. The results were expressed as IC_50_ of NBT reduction.

#### 2.4.4. Hydroxyl Radical Scavenging Test

The extract’s ability to scavenge the hydroxyl radical was performed calorimetrically [23]. An amount of 400 μL FeSO_4_ (1.5 mM), 280 μL H_2_O_2_ (6 mM), 120 μL sodium salicylate (20 mM), and 200 μL of water as blank, or different concentrations of the extract, or the standard (ascorbic acid) in the presence and absence of salicylate (120 μL sodium salicylate (20 mM) or 120 μL water) were incubated at 37 °C for one hour, and the absorbance of hydroxylated salicylate was measured in a Spectrostar Nano plate reader (BMG Labtech) at 562 nm. Based on the results obtained, the percentage of capture of the hydroxyl radical and the IC_50_ was obtained.

#### 2.4.5. Nitrogen Free Radicals Scavenging Test

Nitric acid scavenging activity was estimated by the Griess reaction [22]. Two hundred μL of sodium nitroprusiate (Na_2_[Fe(CN)_5_NO]) 10 mM, 50 μL of PBS, and 50 μL of methanolic extract of *C. sativus* biowaste, or standard (gallic acid and ascorbic acid) diluted in methanol were added into a 96-well plate. Methanol was used as blank. The mixture was incubated protected from light for 150 min at 25 °C. Then, 50 μL of the mixture was added to 100 μL of the sulphanilic acid (4-(H_2_N)C_6_H_4_SO_3_H) reagent (0.33% in 20% glacial acetic acid). After 5 min, 100 μL of 0.1% N-(1-Naphthyl)-ethylenediamine dihydrochloride (C_10_H_7_NHCH_2_CH_2_NH_2_ · 2HCl) were added. It was incubated for 30 min at 25 °C, and finally, absorbance was measured at 540 nm in a Spectrostar Nano plate reader (BGM Labtech). IC_50_ for the extract and the standards (gallic acid and ascorbic acid) were calculated.

### 2.5. Radical Scavenging Activity in Cell Culture (HepG2 and Hs27)

#### 2.5.1. Cell Culture

Human Caucasian hepatocarcinoma cell line HepG2, (Ref. ECACC 85011430) was maintained in EMEM (Eagle’s Minimum Essential Medium) supplemented with 2 mM of glutamine, 1% of non-essential amino acids, 10% of fetal bovine serum (FBS), and 1% of antibiotics (10,000 U of penicillin and 10 mg of streptomycin/mL). This cell line constitutes a human hepatocyte model that has been widely used to evaluate the effects of different compounds of natural origin [24].

Human foreskin fibroblasts cell line Hs27 (Ref. ECACC 94041901) was cultured in DMEM (Dulbeco’s Modified Eagle Medium) supplemented with 2 mM of glutamine, 10% of FBS, and 1% of antibiotics. Cells were incubated at 37 °C under a 5% CO_2_ atmosphere. Hs27 was used as a skin cell model to investigate the effect of the *C. sativa* extract.

For cytotoxicity and ROS assays, cells were resuspended in a medium supplemented with 1% FBS to prevent serum components interaction with the phenolic compounds of the extracts, which may result in artifacts with cytotoxic action [25].

#### 2.5.2. Cytotoxicity

The effect of the extract at different concentrations (500–10 µg/mL) on cell viability following 72 h of treatment was analyzed by the 3-(4,5-dimethylthiazol-2-yl)-2,5-diphenyltetrazolium bromide (MTT) assay [19]. Eight thousand cells/well of each cell line were seeded in a 96-well plate. After 24 h, cells were treated with the extract for 72 h. Then, cells were washed with PBS twice, and 50 µL/well of MTT reagent was added (1 mg/mL MTT in PBS) together with 150 µL of culture medium. After 4 h, the medium was aspirated, and 100 µL/well of DMSO was added to dissolve formazan product. Absorbance was measured at 570 nm in a Spectrostar Nano plate reader (BGM Labtech).

#### 2.5.3. Intracellular ROS Measurement

This assay was conducted to determine the effect of *C. sativus* extract on levels of intracellular ROS. The DCFH-DA assay (2,7-dichloro dihydrofluorescein diacetate) was performed, analyzing two cellular situations [20]. First, the direct effect of the extract was analyzed on HepG2 and Hs27 cells. Cells were seeded at the rate of 8000 cells per well in 150 µL of medium with 1% FBS. After 24 h of incubation, the medium was removed, and 200 μL of DCFH-DA (20 µM) were added. Plates were incubated for 30 min in an incubator. Then, the plates were washed with glucose–PBS (180 mg of glucose/100 mL of PBS), and different concentrations of the extract diluted in EMEM medium enriched with FBS at 1% were added. From the time the extracts were added, fluorescence was measured every 15 min, for a total time of 90 min, at 485 nm of excitation and 520 nm of emission. No fluorescence was detected for the extract diluted in EMEM medium at those wavelengths.

Secondly, the protective effect was analyzed through cells’ pre-treatment with different concentrations of the extract before cell oxidative stress with H_2_O_2_ was induced. Eight thousand cells/well of each cell line were seeded in new plates, and after 24 h of incubation at 37 °C and in 5% CO_2_ atmosphere, they were treated with different concentrations of the extract dissolved in culture medium supplemented with 1% FBS. After another 24 h, the medium was removed, cells washed with glucose-PBS and DCFH-DA added. After 30 min of incubation in darkness, plates were washed with glucose-PBS and the oxidative stress induced (H_2_O_2_, 200 μM). Two controls were prepared, one in the presence of H_2_O_2_ and another without it. Fluorescence was measured every 15 min, for a total time of 90 min from the time hydrogen peroxide was added, at 485 nm of excitation and 520 nm of emission.

### 2.6. Enzymatic Activities

#### 2.6.1. Elastase Inhibition Assay

The assay to determine the ability of the extract to inhibit elastase was based on the hydrolysis measurement of N-suc-(Ala)-3-nitroanilide (SANA) to nitroaniline (colored product) catalyzed by the enzyme [22]. An amount of 200 μL of SANA solution (1 mM) in tris-HCl buffer (0.1 M, pH = 8.0) was mixed with 20 μL of sample (extract at different concentrations), methanol as negative control, or different concentrations of oleanolic acid as a positive control. The mixture was pre-incubated for 10 min at room temperature. Then, 20 μL of the enzyme (0.03 U/mL) were added. The plate was incubated for 10 min, and absorbance was measured in a Spectrostar Nano plate reader (BMG Labtech) at 410 nm.

#### 2.6.2. Hyaluronidase Inhibition Assay

Extract hyaluronidase inhibition capability was conducted following Kim et al. [26] protocol. Oleanolic acid was used as a positive control. Five μL of different concentrations of the extract were preincubated for 10 min at 37 °C with bovine hyaluronidase (1.5 U) dissolved in 100 μL of a solution containing 20 mM sodium phosphate buffer (pH 7.0), 77 mM sodium chloride and 0.01% serum bovine albumin (BSA). The analysis began by adding 100 μL of hyaluronic acid (sodium salt) from rooster comb (0.03% in 300 mM sodium phosphate, pH 5.35). Hyaluronic acid, which had not been digested after 45 min of incubation at 37 °C, was precipitated with 1 mL of an acid albumin solution (0.1% BSA solution in 24 mM sodium acetate and 79 mM acetic acid, pH 3.75). After 10 min at room temperature, absorbance at 600 nm was measured in Spectrostar Nano (BMG Labtech).

#### 2.6.3. Collagenase Inhibition Assay

For this assay, N-[3-(2-furyl) acryloyl]-Leu-Gly-Pro-Ala (FALGPA) was used as collagenase substrate [27]. Methanol was used as a negative control and oleanolic acid as a positive one. Collagenase from *Clostridium histolyticum* (0.8 U/mL) and FALGPA (2 mM) were dissolved in tricine buffer (50 mM tricine, 400 mM NaCl, and 10 mM CaCl_2_, pH 7.5). Before adding the substrate, the enzyme and extract at different concentrations were pre-incubated for 15 min at 25 °C. The final reaction mixture (150 µL) contained tricine buffer, 0.8 mM FALGPA, 0.1 U collagenase, and different amounts of the extract. Collagenase activity was measured at 340 nm every 5 min during 20 min after the substrate was added.

#### 2.6.4. Tyrosinase Inhibition Assay

To determine the effect of the extract on this enzyme, the amount of dopachrome generated as a function of time was measured at 475 nm. α-kojic acid was used as positive control [28]. The reaction was carried out on a plate of 96 wells. Ten μL of the extract at different concentrations or α-kojic acid, or water H_2_O as negative control, 40 μL L-DOPA 5 mM in 63 mM phosphate buffer (pH 6.8), 80 μL of phosphate buffer, and 40 μL tyrosinase 200 U/mL were added in each well. Absorbance was measured every 3 min for a total of 15 min in a Spectrostar Nano plate reader (BGM Labtech) at 37 °C and 475 nm.

#### 2.6.5. Xanthine Oxidase Inhibition Assay

The ability of the extract to inhibit xanthine oxidase was estimated, measuring the amount of uric acid generated, since it corresponds to the final product of the enzyme reaction [20]. An amount of 87.5 μL of phosphate buffer, 10 μL of EDTA (15 mM), 15 μL of xanthin (0.1 mM) 12.5 μL of different concentrations of the extract, or standards (ascorbic or gallic acid), or phosphate buffer as blank, and 25 μL of xanthine oxidase (1 U/10 mL of buffer) were mixed. Measurements were performed at 295 nm at intervals of 5 min for 40 min.

From the obtained percentages of elastase, collagenase, hyaluronidase, tyrosinase, and xanthin oxidase inhibition, the IC_50_ (concentration that inhibits 50% of the enzyme activity) was calculated.

### 2.7. Statistical Analysis

Three independent experiments were conducted in triplicate in all in vitro tests.

Three independent experiments were executed in cell culture assays (8 repetitions per analysis).

After examining homogeneity of variances (Levenne test), ANOVA followed by Bonferroni’s test were performed using the IBM SPSS Statistics 24 program. Statistics with a value of *p* < 0.05 were considered significant and will be indicated in figures with different letters. This means that treatments that do not share a letter were significantly different.

## 3. Results and Discussion

### 3.1. Chemical Analysis of the Extract

#### 3.1.1. Total Phenolic, Flavonoid, and Anthocyanins Content

Results of the extract’s total content in phenols, flavonoids, and anthocyanins are shown in Table 1.

There are many plants which, taken both orally as food supplements or applied topically, are used to reduce photoaging. The secondary metabolites of these plants act mainly as antioxidants. In recent years, plant chemistry has grabbed the attention, and this is mainly due to the growing interest in natural antioxidant sources such as phenols, flavonoids, and/or anthocyanidins [29]. This kind of compound has demonstrated not only its antioxidant and anti-inflammatory activity, but also its capacity to reduce levels or activity of skin enzymes such as those that have been studied in this article [30].

The effectiveness of this type of compound in avoiding wrinkle formation and oxidative damage caused by ultraviolet radiation makes botanical extracts rich in polyphenols an effective and interesting ingredient in the composition of numerous cosmetic creams and extends their therapeutic uses [31]. In view of our results, the total phenol content of saffron tepals (4.59% which is equivalent to 6.50 mg gallic acid equivalents/g of *C. sativus* biowaste) is greater than that found in the skin of some grape varieties which is between 0.57 and 3.71 mg gallic acid equivalents/g [32]. This result allows us to propose *C. sativus* tepals extract as a good candidate for the study of its dermo-protective and skin antiaging capacity.

Flavonoid administration both topically and systemically helps to prevent and treat skin senescence and, therefore, related conditions and diseases. Notably, their wide and safe therapeutic range has been demonstrated in pre-clinical trials [33]. *C. sativus* stigmas can act as antioxidant agents that increase the quality of functional foods, beverages, medicines, and cosmetics [34]. On the other hand, anthocyanins are pigments which give *C. sativus* flowers their characteristic lilac-purple color. Several studies have focused on these compounds’ potential, such as their anti-inflammatory, antimicrobial, antioxidant, free-radical scavenger, antiaging, protection against solar radiation, or anticarcinogenic effects [35].

#### 3.1.2. UHPLC-ESI(±)-QTOF-MS Analysis

In the present study, the UHPLC-ESI(±)-QTOF-MS analysis provided us with an accurate method for a rapid and efficient characterization of the major phenolic compounds from the methanolic extract of *C. sativus* tepals.

Flavonoids were classified by extracting ion chromatograms (EICs) of the characteristic fragment ions of each corresponding aglycone listed in Table 2.

The compounds tentatively identified in the chromatograms are given in Table 2 by their polyphenol classes sequence in each case (http://phenol-explorer.eu; accessed on 19 February 2024), together with their retention times (t_R_ (min)) and MS characteristics. The methanolic extract was rich in flavonols, especially kaempferol and quercetin derivatives, in agreement with the findings of previously described studies [16,36,37].

Following the MS/MS data, 22 different flavonoids and their glycoside derivatives, as well as their corresponding aglycones were traced in the methanolic extract (8 kaempferol derivatives as the most abundant components, 3 hydroxyflavonoids, 5 quercetin and 3 isorhamnetin derivatives and 3 anthocyanins), Table 2. Under the conditions used, most of the phenolic compounds detected had intensive signals corresponding to the deprotonated adduct [M-H]^−^ (Figure 1). However, formation of [M]^+^ was observed especially for anthocyanins [16] (Figure 1). Overall, many different flavonoids were traced in our samples, and the identification of the most abundant one was achieved by comparison to authentic standards when available, with literature structural data [16], and also by studying their retention time and the fragmentation pattern observed that was matched against the predicted MS/MS spectral data (https://foodb.ca, accessed on 19 February 2024).

### 3.2. In Vitro Scavenging Activity

The studied phytoextract has a great diversity of active ingredients, so that the assessment of its antioxidant effect cannot be determined only by one type of test. Each method differs in the way the radicals are generated, the sensitivity of the method, the antioxidant property, or the strategy for measuring the endpoint of the inhibition reaction. It is also essential to use reference substances, such as ascorbic or gallic acid [38].

The scavenging capacity results of DPPH, ABTS, and O_2_^−•^ radicals are shown in Table 3. DPPH and ABTS assays showed tepals extract’s moderate ability, especially if we compare it with the capacity of the analyzed reference substances. The results contrast with those obtained by Sariri et al. [39] for the DPPH radical who reported a IC_50_ of 231.75 μg/mL for a methanolic extract of Iranian saffron floral bio-residues, while for ascorbic acid, they found values similar to ours (20.99 μg/mL). On the other hand, Termentzi and Kokkalou [37] analyze the antioxidant capacity of the saffron flower tepals, extracted with hot methanol (Soxhlet). Their IC_50_ values for DPPH analysis were about double (1700 μg/mL) of those obtained in this work carrying out a cold extraction and preserving anthocyanins by adding HCl. Results indicate that the method chosen in our study is more appropriate than those where higher temperatures are used. However, divergences in results can be also due to climatic variations and to differences in the substrate where the plant grows, the variety used, etc. All these factors affect the concentration and composition of the different active ingredients present in the sample.

Several studies describe a positive correlation between the phenol content and the free-radical scavenging capacity of an extract. In our case, flavonoids and anthocyanins could be mainly responsible for the ability to donate hydrogen atoms of the studied extract. This high content of phenolic compounds explains the greater antioxidant power of saffron flower tepals with respect to their stigmas [40]. Kaempferol, one of the most abundant flavonoids in saffron tepals, has a high antioxidant capacity, but its low solubility in water limits its therapeutic uses. However, the presence of glycosides of this compound in the extract would improve this problem [41].

Quercetin 3-sophoroside has been proposed as primarily responsible for the antioxidant activity of saffron tepals [42]. However, and given the low proportion of this compound in the studied extract (Figure 1, compound 8), we cannot discard a synergistic effect of the antioxidant potential of the anthocyanins, as well as other flavonoids, or the rest of the phenolic compounds present in the sample. In addition, the fact that delphinidin appears in the form of heteroside gives it greater stability, which is essential for its bioavailability [43].

The superoxide radical (O_2_^−•^), in certain processes, can be transformed into the hydroxyl radical. Both ROS play a decisive role in the etiopathogenesis of dermatological conditions such as chronic inflammatory skin diseases [44]. The hydroxyl radical is one of the most reactive and oxidative among ROS, since it is able to initiate a chain reaction that generates damage at the level of genetic material, proteins, and other biomolecules that are essential for cellular survival. It is therefore responsible for aging, mutagenesis, or carcinogenesis [45].

In view of the results represented in Figure 2, we can conclude that the extract of saffron biowaste has a moderate, dose-dependent capacity to capture this radical, since, at the highest studied concentration, the scavenge percentage is near 35%. On the other hand, ascorbic acid scavenged 60% of this radical at the highest concentration studied, with an IC_50_ of 33.9 ± 0.59 μg/mL. Gallic acid, which has been used as a reference substance in other tests, showed no activity to capture this radical.

On the other hand, delphinidin, the most abundant anthocyanin in *C. sativus* extract, is able to scavenge radicals such as hydroxyl, but no NO·[46]. These data are consistent with the results of this study, since no NO·scavenging capacity was detected in the methanolic extract of *C. sativus*, nor in ascorbic acid. This capacity was only observed in gallic acid with an IC_50_ of 386.87 ± 1.76 μg/mL.

### 3.3. Radical Scavenging Activity in Cell Culture (HepG2 and Hs27)

The ability of the extract to capture ROS in cell culture was studied in two cell lines. One hepatocarcinoma, HepG2, was used as a model of human hepatocyte. The liver is responsible for maintaining the metabolic homeostasis of the body, such as the imbalance between free radicals and antioxidant defenses [47,48]. As plants or their extracts can be taken orally to reduce aging due to their antioxidant capacity, the effect of *C. sativus* extract on HepG2 cell line was analyzed. The second Hs27 is a human fibroblasts cell line used as a model to analyze the effect of plant secondary metabolites for skin care [49]. Epidermis and dermis are the two layers that comprise human skin. In this second layer, skin dermal fibroblasts are embedded. Fibroblast senescence due to oxidative stress, mainly because of an increase in ROS levels, has been described during intrinsic and extrinsic skin aging [50]. Fibroblasts are the main components of the dermis and are responsible for producing the components of the extracellular matrix, especially collagen. They are implicated in wound healing and are also responsible for the release of elastic fibers.

In both cases, the effect of the phytoextract was analyzed under two different conditions. The direct effect evaluates the impact of the extract on cells growing under normal culture conditions. The results of this assay are shown in Figure 3. The protective effect allowed us to know the ability of the extract to prevent or to reduce the intracellular ROS concentration in cells growing under oxidative stress induced by H_2_O_2_ [28], and the obtained results are shown in Figure 4.

The results pointed out that the exposure of cell culture to saffron flower extract for 90 min causes a dose-dependent increase in intracellular ROS levels. The effect is much more marked, especially at higher concentrations, on HepG2. In both cases, higher than 66 μg/mL treatments significantly increased intracellular ROS concentrations compared to control. The cytotoxic effect of high concentrations of ROS has been widely documented [51,52]. The overproduction of ROS leads to oxidative stress, which causes cell damage that can be an important mediator of chronic inflammation, which contributes to skin diseases [53]. However, ROS and nitrogen-active species play both beneficial and deleterious roles in cells. Low or moderate concentrations of these radicals act as molecular signals in physiological processes and can enhance immunologic defense [52]. Surprisingly, among them, ROS-mediated actions that protect cells from oxidative stress, helping to maintain their redox homeostasis, can be highlighted [54]. In addition, ROS has a crucial role in the biosynthesis of extracellular matrix components and their crosslinking [55].

Flavonoids are molecules that have demonstrated their antioxidant capacity; however, under certain circumstances, they can also exhibit pro-oxidant activity. This behavior could explain, in part, the observed toxicity of some flavonoids in vivo. However, in practice, the pro-oxidant effects can also be beneficial, since, by imposing a mild degree of oxidative stress, the levels of antioxidant defenses could be elevated, which would lead to greater overall protection of the cell. Although the extent to which flavonoids can act as anti- or pro-oxidants is still partly unknown, it seems to be directly related to their concentration [56]. Our results support this hypothesis since as the concentration of extract increases, so do the levels of ROS. Other studies pointed out that the prooxidant activity of natural antioxidants, including flavonoids and anthocyanins, is catalyzed by metals present in biological systems [57]. The detected ROS increase in both cell cultures, despite being significant, could be considered slight, mainly in the HS27 cell line, and therefore positive for the cell. To ratify this statement, the protective effect of the extract was analyzed.

Hydrogen peroxide induced a significant increase in intracellular ROS levels (stressed control (yellow line)) throughout the 90 min of exposition versus non-stressed control (blue line) in both cell lines, mainly in HepG2 (Figure 4). Pre-treatment with the extract significantly reduces stress levels in a dose-dependent manner. The highest doses of the extract approximate ROS values to those of the control cells, although they do not completely reverse the effect of H_2_O_2_. Therefore, we confirm that the extract can prevent oxidative stress. Related to skin, oxidative stress can be responsible for collagen fragmentation and disorganization, which is related with erythema, rosacea, edema, photoallergic reactions, or psoriasis, but also to cancer development [53].

Polyphenols, including flavonoids and anthocyanins, the major components of the studied extract, show several mechanisms of action through which they exert their antioxidant activity. Among them, the most cited are their ability to scavenge free radicals, especially ROS, or to prevent the progression of the Fenton reaction by the transition metal chelation. In addition, it has been described that these substances are capable of inhibiting ROS-forming enzymes, such as xanthine oxidase, or activate cellular antioxidant enzymes [58]. The mechanism by which polyphenols control the expression of antioxidant enzymes is related with the regulation of PKC activity via Keap1/Nrf2/ARE pathway [59]. Some components of *C. sativus* extract, such as myricetin, isorhamnetin, or delphinidin can activate Nrf2-ARE pathway [60,61].

The chemical complexity of the studied extract would allow the approach of oxidative stress from different mechanisms of action, being the different components able to act synergistically. The free radicals scavenging capacity of the extract explains, at least in part, its chemo-preventive capacity in cell culture under oxidative stress. Polyphenols, both by topical and oral administration, may prevent and alleviate premature skin aging symptoms related with free radicals’ over-production such as those derived from solar radiation and atmospheric pollutants exposition [62].

### 3.4. Dermo-Protective Capacity: In Vitro Enzymatic Inhibition Capacity

#### 3.4.1. Elastase and Collagenase Inhibition Assays

The results pointed out that neither elastase nor collagenase are inhibited by the methanolic extract, since there are no significant differences with the control. New studies for the evaluation of the extract capacity to inhibit metalloproteinases are needed in this respect.

Kanashiro et al. [63] reported that kaempferol, myricetin, and quercetin exert weak inhibitory elastase activity effects, which agrees with our results.

#### 3.4.2. Hyaluronidase Inhibition Assay

Figure 5 shows the results of the effect of saffron biowaste extract and oleanolic acid on the inhibition of this enzyme. It can be observed that the extract has dose-dependent anti-hyaluronidase activity, reaching inhibition levels over 30% at the highest tested concentration. The reference substance, oleanolic acid, is more active than the extract, with an IC_50_ of 0.10 ± 0.001 mg/mL.

Studies conducted in *Vitis rotundifolia* report a correlation between the inhibition of hyaluronidase and total phenol content. The capacity of individual flavonoids to inhibit this enzyme activity including kaempferol, myricetin, and quercetin has been also described [64].

#### 3.4.3. Tyrosinase Inhibition Assay

Tyrosinase is an enzyme present in plants, microorganisms, and animals, involved in melanin synthesis (from L-tyrosine). This substance plays a fundamental role in the defense of the skin against the adverse effects of the sun’s UV rays. Hyperpigmentation derived from excessive activity of the enzyme causes dermatological alterations such as vitiligo, melasma, or lentigo. Currently, inhibitors of this enzyme isolated from plants or microorganisms, such as α-kojic acid, are used as skin-whitening products in medical and cosmetic products [65].

Figure 6 shows the results obtained for the tyrosinase inhibition assay. The extract of saffron biowaste, despite being able to inhibit more than 30% the activity of this enzyme, does not show a dose-dependent effect, showing a similar inhibition in all tested doses.

Extract phenolic compounds, such as anthocyanins and flavonoids, are considered substrates of tyrosinase and constitute the main chemical group of tyrosinase inhibitors by substrate-like interaction. They have also demonstrated their ability to inhibit this enzyme, chelating copper ions (Cu-dependent enzyme). Kubo and Kinst-Hori [66] determined that kaempferol, isolated from the tepals of *C. sativus*, acts as a tyrosinase competitive inhibitor competitive inhibitor (IC_50_ of 67 μg/mL). Meanwhile, 3-O-glycoside derivatives did not inhibit it. The high concentration of kaempferol glycosides in the extract could explain the data obtained in this assay.

#### 3.4.4. Xanthine Oxidase Assay

Xanthine oxidase is a possible target for the prevention of diseases caused by oxidative stress [67]. The overproduction of uric acid mediated by this enzyme is associated with gout and other conditions, such as cardiovascular disorders, nephrolithiasis, or diabetes [68]. The inhibition assay results are shown in Table 4.

As discussed above, the reference substances, both gallic and ascorbic acid, have a greater capacity to capture the superoxide radical produced by this enzyme activity than extracts of saffron bio-residues, since they have lower IC_50_. However, it should be noted that, at the concentrations tested, only the flower extract of *C. sativus* was able to inhibit the enzyme with an IC_50_ of 82.64 μg/mL. It has been shown that xanthine oxidase activity is increased in acne processes (chronic inflammatory condition) contributing significantly to oxidative stress [44]. Therefore, the ability of the extract, not only to capture ROS, but also to inhibit this enzyme, will allow a reduction of oxidative stress responsible for these dermatological alterations.

Among extract components, kaempferol, quercetin, myricetin, and isorhamnetin can inhibit xanthine oxidase activity at low concentrations (IC_50_ values from 0.40 to 5.02 µM), while anthocyanidins have demonstrated less inhibitory capacity [69]. Kaempferol inhibition mechanism is explained by the insertion of this compound into the active site of the enzyme [70].

## 4. Conclusions

It can be concluded that the antioxidant effect of saffron flower tepals, together with its hyaluronidase, tyrosinase, and xanthine oxidase inhibitory capacity, highlights this biowaste as a promising ingredient for cosmetic products. It is especially interesting to prevent damage due to exposure to UVA radiation, because it would help reduce the appearance of spots and damage to the extracellular matrix, preserving its components and therefore skin elasticity, youthfulness, health, and moisture levels. Further studies are necessary to assess its therapeutic value.

## 5. Patents

Acero, N. & Muñoz-Mingarro, D. *Extractos de biorresiduos del azafrán como principios activos de productos cosméticos antioxidantes*. (SPAIN P201630788). Spanish Patent and Trademark Office. 2016.

## Figures and Tables

**Figure 1 antioxidants-13-00358-f001:**
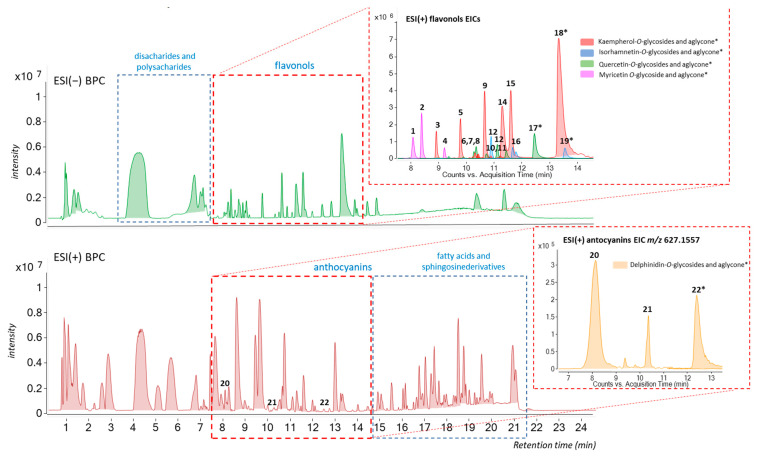
Representative Base Peak Chromatogram (BPC) in positive and negative ion mode ESI(+/−) obtained from LC/MS analysis of the methanolic extract of saffron biowastes (tepals). The numbers with an asterisk (*) correspond with aglycones.

**Figure 2 antioxidants-13-00358-f002:**
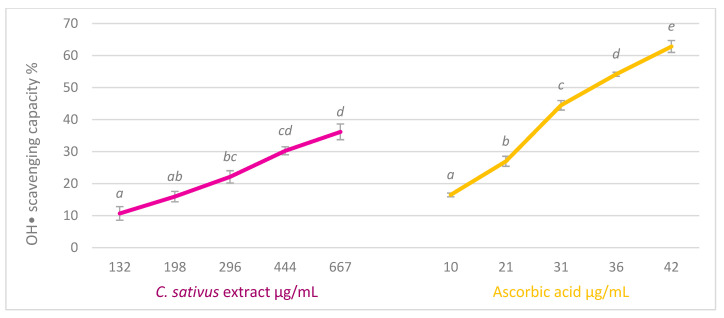
Percentage of hydroxyl radical scavenging capacity of *C. sativus* tepals extract (µg/mL) and ascorbic acid (μg/mL). The results are expressed as mean S.E. of n = 3. Different letters indicate statistically significant differences between concentrations in each treatment (ANOVA Bonferroni, *p* < 0.05).

**Figure 3 antioxidants-13-00358-f003:**
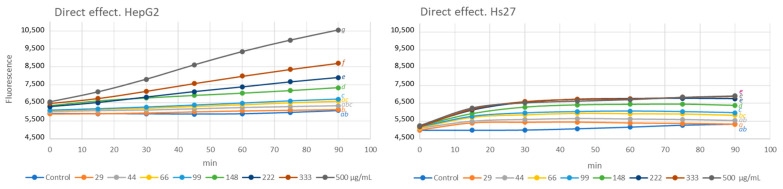
Effect of different concentrations of *C. sativus* tepals extract on HepG2 and Hs27 intracellular ROS levels during 90 min of treatment. Cells growing under normal growth conditions. Different letters indicate statistically significant differences between treatments. ANOVA Bonferroni, *p* < 0.05.

**Figure 4 antioxidants-13-00358-f004:**
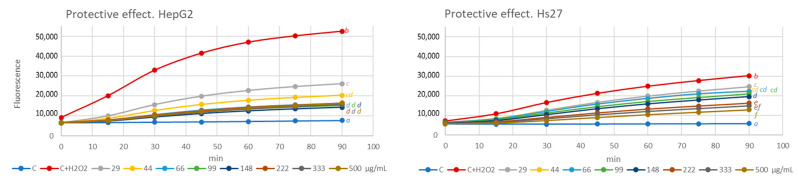
Effect of different concentrations of *C. sativus* tepals methanolic extract on stressed HepG2 and Hs27 intracellular ROS levels during 90 min of treatment. Cells growing under oxidative stress induced with H_2_O_2_ (200 μM). Different letters indicate statistically significant differences between treatments. ANOVA Bonferroni, *p* < 0.05.

**Figure 5 antioxidants-13-00358-f005:**
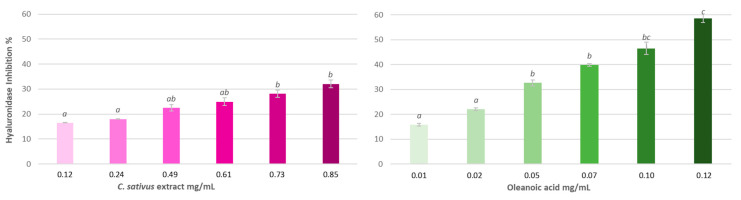
Hyaluronidase inhibition exerted by saffron bio-residue extract and oleanolic acid, expressed as percentage of enzyme inhibition ± S.E. of n = 3. Different letters indicate significant differences (ANOVA, Bonferroni *p* < 0.05).

**Figure 6 antioxidants-13-00358-f006:**
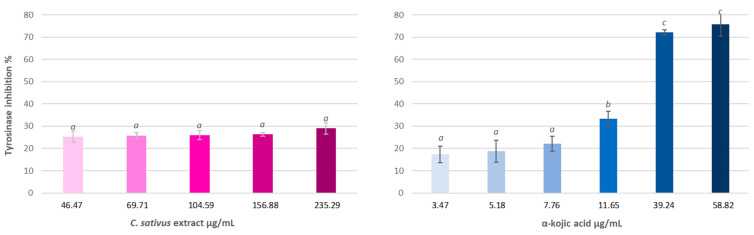
Effect of *C. sativus* extract and α-kojic acid on the activity of the tyrosinase enzyme. The results appear as percentage of enzyme inhibition ± S.E. of n = 3. The different letters indicate significant differences (ANOVA, Bonferroni *p* < 0.05).

**Table 1 antioxidants-13-00358-t001:** Total phenolic, flavonoid, and anthocyanin content of *C. sativus* extract. Total phenolic content is expressed as gallic acid equivalents percentage (*w*/*w*); total flavonoids as epicatechin equivalents percentage; total anthocyanins as delphidin-3-O-glucoside equivalents percentage. The results are expressed as mean ± S.E. of n = 3.

	Total Phenols	Total Flavonoids	Total Anthocyanins
%	4.59 ± 0.09	1.21 ± 0.03	0.68 ± 0.01

**Table 2 antioxidants-13-00358-t002:** Compounds identified by LC-ESI-Q-TOF-MS in *C. sativus* tepals methanol extract.

Nº	Tentative Annotation	t_R_ (min)	[M-H]^−^	MS/MS	Aglycone
	*Flavonols–Kaempherol*				
3	kaempferol 3-*O*-sophoroside-7-*O*-glucoside	8.92	771.2001	[M-Hex(162)-H]^−^ = 609.1468 [M-2Hex(162)-H]^−^ = 446.0850 [M-2Hex(180)-H]^−^= 429.0823	[M-3Hex(162)-H]^−^= 284.0322
5	kaempferol 3,7-di-*O*-glucoside	9.78	609.1471	[M-Hex(162)-H]^−^= 447.0934	[M-2Hex(162)-H]^−^ = 285.0406 [M-2Hex-H-CO]^-^ = 255.0294 151.0035
6	kaempferol 3,7,4’tri-*O*-glucoside	10.32	771.2001	[M-Hex(162)-H]^−^ = 609.1468 [M-2Hex(162)-H]^−^ = 446.0850	[M-3Hex(162)-H]^−^= 284.0322 [M-2Hex-H-CO]^-^ = 255.0294
9	kaempferol 3-*O*-sophoroside	10.65	609.1471	[M-Hex(180)-H]^−^ = 429.0817	[M-2Hex(146)-H]^−^= 284.0329 [M-2Hex-H-CO]^−^= 255.0294 151.0035
11	kaempferol 3-*O*-rutinoside	10.8	593.1506	[M-Hex(146)-H]^−^= 447.0936 [M-Hex-H_2_O-H]^−^= 429.0817	[M-3Hex(162)-H]^−^ = 284.0322 [M-2Hex-H-CO]^-^ = 255.0294
14	kaempferol 3-*O*-glucoside	11.33	447.0941	--	[M-Hex(182)-H]^−^ = 285.0410 [M-2Hex-H-CO]^−^= 257.0457 151.0037
15	kaempferol 7-*O*-glucoside	11.59	447.0941	--	[M-Hex(182)-H]^−^ = 284.0331 [M-2Hex-H-CO]^−^= 255.0301 151.0037
18	kaempferol aglycone	13.39	285.0411	239.0348, 229.0509, 185.0603, 159.0446, 143.0500, 107.0137	
	*3-Hydroxyflavonoids*				
1	myricetin-3-*O*-glucoside	8.10	479.0836	--	[M-Hex(162)-H]^−^ = 317.0302 [M-Hex(162)-H_2_O-H]^−^ = 299.0195 190,9988, 163.0037, 125.0247
2	myricetin-3,7-di-*O*-glucoside	8.41	641.1369	[M-Hex(162)-H]^−^= 479.0836 [M-Hex-H_2_O-H]^−^= 461.0717	[M-2Hex(162)-H]^−^= 317.0302 [M-2Hex(162)-H_2_O-H]^−^ = 299.0195 [M-2Hex-2H_2_O-H]^−^ = 281.0091, 190.9988
4	quercetagetin 3,7-*O*-diglucoside	9.23	641.1369	[M-Hex(162)-H]^−^ = 479.0836 461.0729, 435.0933	[M-2Hex(162)-H]^−^= 317.0302 273.0405
	*Flavonols–Quercetin*				
7	quercetin 3,7-di-*O*-glucoside	10.27	625.1435	[M-Hex(162)-H]^−^ = 463.0882	[M-2Hex(162)-H]^−^ = 301.0355 271.0247
8	quercetin 3-*O*-sophoroside	10.39	625.1435	[M-Hex(162)-H]^−^ = 463.0882	[M-2Hex(162)-H]^−^= 301.0355 271.0247
10	quercetin 3-*O*-glucoside	11.15	463.0902	--	[M-Hex(162)-H]^−^= 301.0350 271.0247, 255.0296
13	quercetin 7-*O*-glucoside	11.57	463.0902	--	[M-Hex(162)-H]^−^= 301.0355 273.0409, 255.0297
17	quercetin aglycone	12.44	301.0359	273.0402, 178.9981, 151.0035, 121.0293, 107.0139	
	*Flavonols–Isorhamnetin (7-methoxiquercetin)*				
12	isorhamnetin 3-*O*-glucoside-7-*O*-rhamnoside	10.90	623.1628	[M-Hex(164)-H]^−^= 459.0926	[M-2Hex(145)-H]^−^ = 314.0432 [M-3Hex-CH_3_-H]^−^= 299.0195 285.04105, 271.0250, 151.0024
16	isorhamnetin 3-*O*-glucoside	11.79	477.1031	--	[M-Hex(162)-H]^−^ = 314.0432 [M-2Hex-CH_3_-H]^−^ = 299.0197 285.04105, 271.0236
19	isorhamnetin aglycone	13.56	315.0519	[M-CH_3_-H]^−^ = 300.0280, 283.0255.271.0250, 255.0310	
Nº	Tentative Annotation	t_R_ (min)	[M]^+^	MS/MS	aglycone
	Anthocyanins				
20	delphinidin 3,5-di-*O*-glucoside	8.0	627.1550	[M-Hex(162)]^+^ = 465.1024	[M-2Hex(162)]^+^ = 303.0490
21	delphinidin 3-*O*-sophoroside	10.20	627.1550	[M-Hex(162)]^+^ = 465.1022	[M-2Hex(162)]^+^ = 303.0500
22	delphinidin aglycone	12.46	303.0492	285.0392, 257.0440, 229.0487, 201.0537	

**Table 3 antioxidants-13-00358-t003:** DPPH, ABTS, and O_2_^−•^ scavenging activity (IC_50_ µg/mL) of saffron tepals extract and reference substances: ascorbic and gallic acid. The results are expressed as mean ± S.E. of n = 3.

Radical	*C. sativus* Extract IC_50_	Ascorbic Acid IC_50_	Gallic Acid IC_50_
DPPH	887.87 ± 10.39 µg/mL	21.18 ± 0.21 µg/mL	0.803 ± 0.001 µg/mL
ABTS	18.4 ± 0.22 µg/mL	0.31 ± 0.01 µg/mL	0.91 ± 0.02 µg/mL
O_2_^−^•	58.46 ± 2.71 µg/mL	11.64 ± 0.30 µg/mL	2.30 ± 0.05 µg/mL

**Table 4 antioxidants-13-00358-t004:** Xanthine oxidase inhibition capacity (IC_50_ µg/mL) of saffron tepals extract and reference substances: ascorbic and gallic acid. The results are expressed as mean ± S.E. of n = 3.

O_2_^−^	*C. sativus* Extract	Gallic Acid	Ascorbic Acid
IC_50_ (µg/mL)	82.64 ± 3.32	-	-

## Data Availability

The raw data supporting the conclusions of this article will be made available by the authors on request.

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
