# Peer review of "Effects of Crocus sativus L. Floral Bio-Residues Related to Skin Protection"

_antioxidants, 2024, doi:10.3390/antiox13030358_

Round 1

Reviewer 1 Report

It's an interesting work focusing on the skin antiaging properties of Crocus sativus L.However there are some points that need to be clarified before acceptance

The main point is that the authors claim the use of hepG2 and hs27 cell lines so as to assess dermo-protective effect of a complex methanolic extract of C. sativus tepals. Why the authors use these types of cell lines instead of using primary human fibroblasts or human keratinocytes? These primary cell lines simulate better the function of human dermis and epidermis. The authors should explain their choice of cells in the text, either in introduction or in materials and methods part.

In the materials and methods part the authors should add how many independent experiments were executed as long as the number of repetitions per analysis.

the figures are very clear and present well the results.

Author Response

We warmly thank the reviewer for the time spent in revising our paper. We have addressed his/her comments and hope that the revised version will be satisfactory. The Reviewer's comments are reproduced in the attached file, with our responses following each comment.

Reviewer 2 Report

The manuscript entitled "Skin anti-aging effect of Crocus sativus L. floral bio-residues" describes the chemical and biological analysis of a methanolic extract of C. sativus tepals, attempting to support its use as a cosmetic ingredient. It is a well-designed, meticulous study with interesting results. However, some revisions are necessary to make the manuscript suitable for publication in Antioxidants:

1) A more precise title should be applied. There is no documentation of direct anti-aging effects of the extract in the manuscript. I would suggest a more precise title, e.g., "Effects of Crocus sativus L. floral bio-residues related to skin protection".

2) In Figure 3, a pro-oxidant effect of the extract is shown, which comes in sharp contrast to the antioxidant effects shown in Figure 4. This may be due to the different experimental design between the two experiments. In particular, in the experiments shown in Figure 3, the extract was present in the culture medium during fluorescence measurements (lines 204-205), while in the ones described in Figure 4, after pre-treatment with the extract, fluorescence measurements were performed in the absence of the extract (lines 206-215). The authors should perform experiments studying ROS basal levels after extract removal (as an enrichment of Figure 3), as well as, studying the suppression of H2O2-induced ROS levels in the presence of the extract, i.e. adding extract along with H2O2 (as an enrichment of Figure 4). This will help to elucidate the discrepancy between the two experimental results.

In both Figures 3 and 4, a positive control is also necessary (e.g. the water-soluble analogue of vitamin E, Trolox), as the authors have done in all other experiments. Moreover, in both figures the authors should include each time-point, instead of showing only the line connecting the time-points.

There are several language errors. I have selected some of them, but the authors should go through the whole manuscript checking carefully for more errors:

In line 55 should read "their" instead of "they"

In line 128 should read "in" instead of "by"

In line 192 should read "...on cell viability following 72 h..."

In line 206 should read "through cells' pre-treatment" instead of "trough cell pre-treatment"

In line 292 should read "identified" instead of "identify"

In line 304 should read "abundant" instead of "abundance"

In line 395 should read "implicated" instead of "implied"

In line 474 should read "Hyaluronidase" instead of "Hialuronidase"

Author Response

(The authors gave the same response as above.)

Reviewer 3 Report

- Material and Methods, subparagraph 2.3.1. (lines 111-118) – Material and methods, subparagraph 2.4. (lines 158 – 175) – Material and methods, subparagraph 2.5.2 (lines 192 – 194) – Material and methods, subparagraph 2.6. (lines 218 – 239): Although bibliographical references have been included for these assays, it should be better to specify the details of these assays to allow the readers to repeat the experimental procedures (e.g., the procedure, the reagents volumes, the linearity range of standards calibration curves, the solubilization solvents of the extracts, the limits of detections and quantifications).

- Results and discussion, subparagraph 3.1.1.: The authors reported the total phenol content as % in Table 1. Instead, on line 268, they reported it as mg gallic acid equivalent/g of C. sativus biowaste. Biowaste results are referred to the extracts or the freeze-dried matrix? In addition, it should be more appropriate to explain the results in Table 1 and specify in Material and Methods the calculation of the %.

- Figures 3 and 4: It should be better to improve the quality and the clarity of the figures. Different colors are recommended to distinguish the C. sativus tepals extract concentrations.

- Abstract, line 16: Please, add “)” after “(UHPLC-ESI-QTOF-MS”.

Author Response

(The authors gave the same response as above.)

Round 2

Reviewer 2 Report

The revised version of the manuscript is now suitable for publication in Antioxidants.

There are no other comments.